# Spatial Effect of Industrial Energy Consumption Structure and Transportation on Haze Pollution in Beijing-Tianjin-Hebei Region

**DOI:** 10.3390/ijerph17155610

**Published:** 2020-08-04

**Authors:** Meicun Li, Chunmei Mao

**Affiliations:** 1Business School, Hohai University, Nanjing 211100, China; 2Centre for Environmental Policy, Imperial College London, South Kensington, London SW7 1NA, UK; 3School of Public Administration, Hohai University, Nanjing 211100, China; maochm@hhu.edu.cn

**Keywords:** PM_2.5_, Beijing-Tianjin-Hebei, industrial energy consumption structure, transportation, spatial econometrics

## Abstract

Haze pollution has a serious impact on China’s economic development and people’s livelihood. We used data on PM_2.5_ concentration, industrial energy consumption structure, economic development and transportation in Beijing-Tianjin-Hebei and surrounding cities from 2000 to 2017, and analyzed the spatial effect of industrial energy consumption structure and traffic factors on haze pollution by using spatial autoregressive model (SAR) and spatial error model (SEM). The results indicated that: (1) The global spatial correlation analysis showed that haze pollution had a significant positive spatial correlation, and the local spatial correlation analysis showed that the high-high clusters of PM_2.5_ were located in the south and middle of the region; (2) The change of industrial energy consumption structure was highly correlated with haze pollution, namely, the increase of industrial energy consumption led to the deterioration of environmental quality; (3) The change of economic development was highly correlated with haze pollution. There was no clear EKC relationship between haze pollution and economic development in Beijing-Tianjin-Hebei region and surrounding cities. However, the relationship was similar to inverted U-shaped curve; (4) The change of traffic jam was highly correlated with haze pollution, namely, the increase of fuel consumption per unit road area led to the deterioration of environmental quality. Based on the above results, from the perspective of space, the long-term measures for haze control in Beijing-Tianjin-Hebei and surrounding cities can be explored from the aspects of energy conservation and emission reduction, industrial transfer, vehicle emission control, traffic restrictions and purchase restrictions.

## 1. Introduction

A wide range of haze has covered central and eastern China in recent years. Haze pollution has brought serious health effects to China, and its economic cost has been rising year by year [1]. Chinese government highly values ecological and environmental protection. Guided by the conviction that lucid waters and lush mountains are invaluable assets, the country advocates harmonious coexistence between humans and nature, and sticks to the path of green and sustainable development [2] (pp. 279–285). In 2013, the Chinese government issued the implementation of action plan for the control of air pollution in the Beijing-Tianjin-Hebei region and surrounding areas. The aim was to lower air pollution in the region through joint control of regional air pollution [3].

Spatial econometric panel data model was developed by the cross-section data of multiple time nodes. Anselin [4] introduced spatial effect into the cross-section data model, and proposed Spatial Autoregression Model (SAR) and Spatial Error Model (SEM). Anselin et al. [5] proposed the Ordinary Least Squares (OLS) for the benchmark models of SAR and SEM, and developed the Lagrangian Multiplier (LM) statistic to test the autocorrelation of the spatial lag term and the spatial error term. Using spatial statistical models with fixed effects items and random effects items, spatial econometric panel data model controls the individual heterogeneity.

Space effect, coal burning, industry and automobile exhaust are important factors contributing to the rise of PM_2.5_ concentration [6]. The impact of coal burning and industrial production on pollution is essentially related to the industrial energy consumption structure [7], while automobile exhaust belongs to the transportation sector [8]. The distribution of PM_2.5_ has unique spatial and temporal characteristics [9,10], and its transmission characteristics within and between regions have significant positive correlation [11]. This study combines the three factors of industrial energy consumption structure, economic development and transportation to discuss the haze problem [12,13,14,15,16], and attempts to analyze the problem from the perspective of space. 

For the causes of haze pollution, scholars, especially natural science researchers, have conducted in-depth component analysis, tracking and monitoring. Anselin [17], Gray and Shadbegian [18], Bateman et al. [19] specifically discussed the importance of spatial factors to environmental economy research. Rupasingha et al. [20], Maddison [21] and Chen et al. [22] used the spatial econometric method to discuss the relationship between per capita income and air pollution, and the results showed that the use of spatial variables greatly improved the accuracy of the measurement model. Xue and Geng [11] using data from the China National Environmental Monitoring Centre (CNEMC) found that PM_2.5_ transmits across regions. Hosseini and Kaneko [23] used SDM to study the atmospheric conditions of 129 countries, and proved that pollutants could spread to neighboring countries, and a significant inverted U-shaped EKC curve could be seen after controlling the spatial relationship. Ding et al. [24] adopted SDM to test whether there was EKC trend between PM_2.5_ and economic growth in Beijing-Tianjin-Hebei region, and the results showed that economic growth and PM_2.5_ presented an obvious inverted U-shaped EKC curve. Compared with the non-spatial model, the turning point of EKC is more likely to occur when considering the spatial effect. Ma et al. [25] used SEM and SAR to analyze the interaction effect of inter-provincial PM_10_ and the influence of industrial structure, and found that industrial transfer lacked long-term impact on reducing pollution. Taking Beijing as an example, they found that traffic jam and spatial factors were important reasons for severe pollution.

Pollution control policies should not only be analyzed in a single region, but also in urban agglomeration or the whole region [26,27] (pp. 123–155). Chen et al. [1], Adgate et al. [28], Liu et al. [29] and Bell et al. [30] pointed out the distribution of PM_2.5_ is spatiotemporal heterogeneity. 

Although the methods used in various studies are different, and limited by the variability of PM_2.5_ components and subjective factors of researchers, it can be concluded that PM_2.5_ sources in key Chinese cities mainly include three aspects: coal burning sources, industrial sources and automobile exhaust, of which the contribution of automobile exhaust is about 10% to 30% [9]. Poon et al. [6], Han and Hayashi [31] and Wu et al. [32] pointed out that although the private car ownership rate in China is lower than that in the United States and other western countries, with the increase of per capita income, this rate will keep growing and pose a serious threat to China’s environment, especially the urban environment. Zheng and Huo [33], Barth and Boriboonsomsin [34,35], Greenwood et al. [36] and Jerrett et al. [37] illustrated the connection between urban traffic congestion and carbon emissions from a microscopic perspective, and discuss the importance of urban spatial structure for easing congestion and improving the environment. In previous studies, Chinese scholars pay more attention to the impact of energy structure and economic development on the environment. Research on the impact of transportation on China’s environment started late and focused mainly on carbon emissions. Most of the studies used static panel model, ignoring the mechanism of spatial interaction. Referring to the existing research experience and combining with the spatial measurement method, this study takes the newly defined heavily polluted area (Beijing-Tianjin-Hebei and surrounding cities) as the research object, and uses the spatial panel data from 2000 to 2017 to analyze the influencing factors of PM_2.5_ from three aspects: industrial energy consumption structure, economic development level and traffic jam.

## 2. Materials and Methods

### 2.1. Study Area

The area to be studied is the Beijing-Tianjin-Hebei region and its surrounding areas, a total of 28 cities. These cities are among the most polluted areas in China. The Chinese government proposed the concept of ‘26 + 2’ cities in 2017, which means that 26 cities, together with Beijing and Tianjin, constitute the key areas of air pollution control in China. The location of the region in China is highlighted in Figure 1.

### 2.2. Global Spatial Correlation

Tobler’s First Law of Geography stated that ‘Everything is related to everything else, but near things are more related to each other.’ [38]. On this basis, spatial econometrics abandoned the traditional null hypothesis that the space does not matter for economic relationships, and added spatial dimension to the econometric model to analyze the data more accurately.

Due to the existence of Tobler’s First Law, a large number of literatures focus on the spatial correlation between adjacent regions. The spatial correlation of PM_2.5_ can be determined by measuring the global Moran’s I index. The formula is as follows:(1)$$Μοran’s\;I=\frac{\sum_{i=1}^{n}\sum_{j=1}^{n}w_{ij}(x_{i}-\bar{x})(x_{j}-\bar{x})}{S^{2}\sum_{i=1}^{n}\sum_{j=1}^{n}w_{ij}},\;S^{2}=\frac{\sum_{i=1}^{n}{\left(x_{i}-\bar{x}\right)}^{2}}{n}$$
where $w_{ij}$ is the spatial weight matrix; $x_{i}$ and $x_{j}$ is the observed value of the space units; n is the number of cities in the study region; $\bar{x}$ is the average value; $S^{2}$ is the variance. Moran’s I is between −1 and 1. When Moran’s I is greater than 0, it means there is a positive spatial correlation between space units; when it is equal to 0, it means that there is no spatial correlation; when it is less than 0, it means that there is a negative spatial correlation. The setting principle of w is: (2)$$w_{ij}=\{\begin{array}{l} 1\;when\;region\;i\;is\;adjacent\;to\;region\;j\\ 0\;when\;region\;i\;is\;not\;adjacent\;to\;region\;j\\ 0\;when\;i=j \end{array}$$
where ‘adjacent’ includes diagonal adjacent. In other words, as long as two regions have a common boundary or intersection point, they are defined as ‘adjacent’.

### 2.3. Local Spatial Correlation

The global Moran’s I reflects the overall spatial correlation. Anselin [39] pointed out that the overall evaluation may ignore the atypical characteristics of local areas. It is necessary to use the local correlation index (LISA) to evaluate the specific correlation of local areas and whether there is a significant spatial cluster, which can be tested by measuring the local Moran’s I. The formula is as follows:(3)$$Moran’s\;I_{i}=\frac{\left(x_{i}-\bar{x}\right)}{S^{2}}\sum_{j=1}^{n}w_{ij}(x_{j}-\bar{x})$$
where $Moran’s\;I_{i}$ measures the correlation of PM_2.5_ between region i and its surrounding areas. When $Moran’s\;I_{i}$ is greater than 0, it means that regions with similar values cluster together, which is manifested as high-high or low-low spatial clusters. When $Moran’s\;I_{i}$ is less than 0, it means that the regions with different values gather together, which is manifested as high-low or low-high spatial outliers.

### 2.4. Spatial Econometric Panel Data Model

The spatial econometric panel data model of the subject can be established after verifying that the outcome has spatial correlation. Spatial Autoregressive Model (SAR) and Spatial Error Model (SEM) are established to analyze the spatial effects of industrial energy consumption structure, economic development and traffic factors for PM_2.5_. Based on the traditional econometric model, a spatial autoregressive term is added to construct a spatial autoregressive model to analyze the spatial spillover effects of variables. The model is defined as follows:(4)$$\begin{array}{l} \ln PM_{it}=\alpha_{1}ES_{it}+\alpha_{2}\ln GDP_{it}+\alpha_{3}TJ_{it}+\rho w_{ij}\ln PM_{it}+\delta_{it}+\upsilon_{it}+\varepsilon_{it},\\ \varepsilon_{it}\sim N(0,\sigma_{it}^{2}) \end{array}$$
where ln *PM_it_* refers to the concentration of PM_2.5_ in year t of city i; $GDP_{it}$ is the GDP in year t of city *i*; $ES_{it}$ is the measurement index of industrial energy consumption structure in year t of city i; $TJ_{it}$ is the measurement index of transportation in year t of city i; ln is the logarithm of the variable; $\delta_{it}$ is the time effect; $\upsilon_{it}$ is the individual effect; $\varepsilon_{it}$ is the disturbance term. ρ is the coefficient of spatial variable and represents the degree of spatial spillover effect. $\rho w_{ij}\ln PM_{it}$ refers to the relationship of PM_2.5_ between one city and its neighboring cities.

The spatial error model indicates that the spatial spillover effect results from the disturbance term. The model is defined as follows:(5)$$\begin{array}{l} \ln PM_{it}=\alpha_{1}ES_{it}+\alpha_{2}\ln GDP_{it}+\alpha_{3}TJ_{it}+\delta_{it}+\upsilon_{it}+\varepsilon_{it},\\ \varepsilon_{it}=\lambda w_{ij}\varepsilon_{it}+\mu_{it},\mu_{it}\sim N(0,\sigma_{it}^{2}) \end{array}$$
where λ is the spatial variable coefficient of the disturbance term. The meanings of other symbols are consistent with those in the spatial autoregressive model.

### 2.5. Data Sources and Processing

The outcome variable is the logarithm of PM_2.5_ concentration in each city from 2000 to 2017. Data are drawn from the Atmospheric Composition Analysis Group [40] at Dalhousie University uses NASA satellites and ground stations to monitor and record global PM_2.5_ concentration. The publicly available original data are raster data with a resolution of 0.01° × 0.01°. This paper collected the PM_2.5_ concentration of ‘26 + 2’ cities in Beijing-Tianjin-Hebei region and surrounding areas from 2000 to 2017. In order to reduce the interference of heteroscedasticity, the values of PM_2.5_ concentration was logarithmically processed to form panel data for spatial correlation analysis.

The explanatory variables were industrial energy consumption structure (ES), economic development level (lnGDP) and traffic jam (TJ). 

The structure of industrial energy consumption is the ratio of output value of high-consumption coal industry to GDP. This study collected the industrial output data of eight energy-intensive industries, which included the production and supply of electricity and heat, petroleum processing, coking and nuclear fuel processing industry, ferrous metal smelting and rolling industry, non-metallic mineral products industry, mining and washing of coal, chemical raw materials and chemical products manufacturing, nonferrous metal smelting and rolling industry, and papermaking and paper products [25]. The industrial output of eight energy-intensive industries were added up, and the ratio of them to the current year’s GDP was calculated. Data were obtained from the Statistical Yearbook (2001–2018) [41,42,43,44,45] of 28 cities and China Industrial Economy Statistical Yearbook (2001–2018) [46]. 

The transportation variable captures the degree of traffic jam. This is defined as the ratio of the consumption of gasoline and diesel oil of urban residents to the urban road area (t/m^2^). In 2006, the Intergovernmental Panel on Climate Change (IPCC) published the relationship between the speed of private cars and the amount of petrol consumed per mile in the Guidelines for National Greenhouse Gas Inventory [47]. It pointed out that the slower a car goes, the more petrol it consumes. In traffic jams, the petrol consumption is almost twice that of normal driving. In recent years, the rapid increase of the consumption of gasoline and diesel oil is partly due to the increasing rate of private car ownership. On the other hand, serious urban traffic congestion is also an important cause. This paper attempts to use the consumption of gasoline and diesel oil of urban residents to measure the degree of traffic jam. The higher the fuel consumption per unit road area, the higher the degree of traffic jam. Ma et al. [25] took traffic congestion as an independent variable and carried out a spatial analysis of the influencing factors of haze pollution, and found that the indicator of traffic congestion was not significant in whole China. This study made a spatial analysis of Beijing-Tianjin-Hebei region and surrounding cities to further explore the impact of traffic jam on haze pollution. This paper has also tried to use traffic pressure, namely the ratio of the number of private car ownership to the length of regional highways, as a traffic factor for spatial regression analysis, but the regression result is not ideal. There is a high autocorrelation between traffic pressure and traffic jam. After comparing the two regression results, traffic jam was selected as the explanatory variable representing the traffic factor. Data were obtained from the Statistical Yearbook (2001–2018) [41,42,43,44,45] of 28 cities and the Statistical Communique on National Economic and Social Development (2000–2017) [48] of 28 cities.

## 3. Results

### 3.1. Spatial Correlation Analysis of Haze Pollution

Table 1 reveals the results of the global Moran’s I for PM_2.5_ in Beijing-Tianjin-Hebei region and surrounding cities. As shown in Table 1, the global Moran’s I were highly significant at *p* < 0.01. For a city with high/low PM_2.5_ concentration, there was at least one city with high/low PM_2.5_ concentration adjacent to it. From 2000 to 2017, this positive spatial correlation fluctuated between 0.3–0.5 and reached its peak in 2007, indicating that this positive spatial correlation developed steadily during this period. Moran scatter plot of PM_2.5_ in Beijing-Tianjin-Hebei region and surrounding cities is shown in Figure 2. The horizontal axis represents the standardized PM_2.5_ concentration value, and the vertical axis represents the spatial lag value of the standardized PM_2.5_ concentration value. The scatter plot takes the average value as the axis center, and the first and third quadrants respectively represent the high-high and low-low spatial clusters. The global Moran’s I is positively correlated, which means that the second and fourth quadrants of high-low and low-high spatial outliers are atypical observation areas. From 2000 to 2017, apart from 4–8 cities in the second and fourth quadrants, most cities were in the typical observation area, which further indicated that the spatial positive correlation of PM_2.5_ had a strong stability [49].

The local Moran’s I statistic was suggested in Anselin [39] as a way to identify local spatial clusters and local spatial outliers. Figure 3 reveals the results of the cluster map for PM_2.5_ in Beijing-Tianjin-Hebei region and surrounding cities in 2000, 2008 and 2017. The local Moran’s I were significant at *p* < 0.05. As shown in Figure 3, the low-low spatial cluster was located in the west of the region (Taiyuan and Yangquan). In 2000, the high-high spatial clusters were located in the south of the region (Zhengzhou, Kaifeng, Xinxiang, Hebi, Puyang and Heze), and gradually moved to the middle of the region (xingtai, Dezhou and Cangzhou). From 2000 to 2017, the frequencies of high-high spatial cluster in Xingtai, Zhengzhou, Dezhou, Heze and Kaifeng were all higher than 7, while the frequencies of low-low spatial cluster in Taiyuan and Yangquan were higher than 10. According to the results, the high pollution concentration areas in Beijing-Tianjin-Hebei region and surrounding areas were mainly located in the south and the central part, and showed a trend of transferring from the south to the central part.

### 3.2. Regression Results and Analysis of Spatial Effects

The results of the global Moran’s I showed that PM_2.5_ concentration had a significant positive spatial correlation in Beijing-Tianjin-Hebei region and surrounding cities. To account for this spatial econometric panel data model was estimated to analyze the influence of industrial energy consumption structure, economic development and traffic jam on PM_2.5_. Table 2 showed the results of Lagrange Multiplier (LM) of spatial autoregression model and spatial error model, and further compared the suitability of these two models. The results showed that LM(lag) and LM(error) in SAR and SEM models were highly significant at *p* < 0.001. Further observation of the results of R-LM(lag) and R-LM(error) showed that R-LM(error) was significant at *p* < 0.001, but R-LM(lag) was not significant, indicating that the spatial error model was more suitable for this study than the spatial autoregression model. The Hausman test result showed that the statistical value of Hausman test was 83.75. The null hypothesis of random effect was rejected at *p* < 0.001. It is obvious that the fixed effect model should be selected for analysis. Compared with other models, the value of R^2^ in model(4) showed that the model had a low fitting degree, so model(5) and model(6) were selected as the reference models for analysis. The values of λ in model(5) and model(6) were both greater than 0, indicating that PM_2.5_ had spatial spillover effect. According to model(6), for every 1% increased in PM_2.5_ in surrounding cities, the PM_2.5_ in the central city increased by 0.651%. The spatial spillover effect was obvious.

## 4. Discussion

### 4.1. Influence of Industrial Energy Consumption Structure on Haze Pollution

According to models (5) and (6) in Table 2, the variable of industrial energy consumption structure was significant at *p* < 0.001. The change of industrial energy consumption structure was highly correlated with haze pollution. For every 1% increase in ES in the region, PM_2.5_ increased by 0.131% or 0.134%.

Energy conservation and emission reduction policies have an impact on PM_2.5_. From 2000 to 2007, the PM_2.5_ concentration in Beijing-Tianjin-Hebei region was on the rise. In 2003, China stepped into the market-oriented heavy industrialization stage, and heavy industry was the ‘big consumer’ of coal [4]. As a result, coal consumption accounted for an increasing proportion of total energy consumption, which was an important reason for the rising trend of PM_2.5_. From 2008, the central and local governments began to implement a series of energy conservation and emission reduction policies. For example, in 2013, the Ministry of Environmental Protection set a target of reducing PM_2.5_ concentration by 25 percent by 2017 in Beijing-Tianjin-Hebei region, compared with 2012. In 2014, the Beijing-Tianjin-Hebei region jointly set the goals to give top priority to environmental protection in the integration process and gradually realize unified emission standards. In 2015, Beijing stipulated that consumers across the city should be encouraged to buy energy-saving products, and eligible consumers should be given consumption subsidies. From 2008 to 2017, in the process of promoting the transfer of investment from high-polluting industries to low-polluting industries, the proportion of coal in the energy consumption structure of all cities decreased.

Industrial transfer affects the regional industrial structure and further affects the concentration of PM_2.5_. As shown in Figure 3, the high-high spatial cluster of PM_2.5_ were mostly in the south and central regions. In the process of industrial structure transfer in Beijing-Tianjin-Hebei region, a development pattern gradually formed from ‘Beijing and Tianjin as the exporter, Hebei, Shanxi and other neighboring cities as the importer’ to ‘Beijing-Tianjin-Hebei region as the exporter, central and western China and northeast China as the importer’. These industries were mainly resource-based, and most of them were high-consumption and high-pollution industries. When the capacity of pollution control remains unchanged or the speed of capacity improvement is far lower than that of industrial cluster, the positive externalities of economic development and the negative externalities of pollution concentration will occur together.

### 4.2. Influence of Economic Development on Haze Pollution

According to models (5) and (6) in Table 2, the variable of lnGDP was significant at *p* < 0.001. The change of economic development was highly correlated with haze pollution. For every 1% increase in lnGDP in the region, PM_2.5_ increased by 0.057% or 0.055%. Due to the existence of Environmental Kuznets Curve (EKC), the impact of economic development on haze pollution needs to be further analyzed.

EKC emphasizes that in the early stages of a country’s economic development, pollution levels rise as per capita income increases. When economic development and income reach a certain level, the further growth of income will lead to the improvement of environmental quality or the reduction of pollution. There is an inverted U-shaped relationship between the variation trend of pollutants and per capita income. Given the difficulty of collecting data, most studies used GDP instead of income. Figure 4 reveals the results of the relationship between lnGDP and lnPM_2.5_ in Beijing-Tianjin-Hebei region and surrounding cities. From 2000 to 2006, with the development of economy, the haze pollution level showed a fluctuating rising trend, which reached the peak in 2006 and then showed a fluctuating decreasing trend, indicating that there was no clear EKC relationship between haze pollution and economic development in Beijing-Tianjin-Hebei region and surrounding cities, but the relationship was similar to inverted U-shaped curve. At present, with the development of the economy, the level of haze pollution shows a fluctuating decreasing trend.

### 4.3. Influence of Transportation on Haze Pollution

According to models (5) and (6) in Table 2, the variable of transportation was significant at *p* < 0.001. The change of traffic jam was highly correlated with haze pollution. For every 1% increase in TJ in the region, PM_2.5_ increased by 2.637% or 2.700%.

In China, the problem of traffic congestion seriously affects the development of urban environment. The more developed countries are, the more significant the impact of transportation on environmental quality will be, while for the less developed countries, energy structure is a more significant factor. According to the Energy Information Administration (EIA), transportation is the main factor affecting air quality in developed countries such as the United States, Canada and Australia, while energy structure is the dominant factor in developing countries such as India and South Africa. With the rapid development of regional economy, transportation has become an important factor affecting urban environment.

Regional motor vehicle emission control policies have impacts on haze pollution. From 2003 to 2008, Beijing implemented stage 2, stage 3 and stage 4 of the national emission standard, which is the same as European standard. At present, the government encourages the public to use vehicles with stage 5 of national emission standard or clean energy and new energy vehicles. In 2009, Handan, Taiyuan and Jinan issued regulations on the control of motor vehicle exhaust. The qualification rate of environmental protection test for motor vehicles was more than 90% in Tianjin, Hengshui, Taiyuan, Changzhi, Jinan and Binzhou. Strict implementation of the national emission standard is conducive to the regional control of PM_2.5_.

Traffic restrictions and purchase restrictions of motor vehicle have impacts on haze pollution. In 2014, Beijing, Tianjin and Shijiazhuang began to implement traffic restrictions. At present, Xingtai, Handan, Taiyuan, Yangquan, Jincheng and Changzhi have also implemented strict traffic restrictions. However, some studies showed that traffic restrictions have a positive effect on haze control in the short term, but the effect can be weakened in the long term. Beijing, Tianjin and Shijiazhuang have implemented purchase restrictions of motor vehicle, and the growth rate of total vehicle ownership has been controlled. However, in 2018, China’s auto market showed negative growth for the first time in 28 years, and the auto demand had fallen for three years. In contrast to the negative automobile consumption, large cities, such as Beijing, have been implementing traffic restrictions and purchase restrictions for a long time, resulting in a large amount of pent-up consumption demand.

## 5. Conclusions

Based on the panel data of PM_2.5_ in Beijing-Tianjin-Hebei region and surrounding cities, this paper constructed the spatial autoregressive model and spatial error model to study the influence mechanism of PM_2.5_ from three aspects: industrial energy consumption structure, economic development and transportation.

The study found that PM_2.5_ in Beijing-Tianjin-Hebei region and surrounding cities had significant spatial cluster, and the change of PM_2.5_ in one city formed spatial spillover effect on adjacent cities. Industrial energy consumption structure, economic development and transportation are the significant factors influencing PM_2.5_. The industrial energy structure was analyzed from two aspects of energy conservation and emission reduction, industrial transfer. Through EKC, the influence of economic development was further analyzed. The factor of traffic jam was analyzed from three aspects of vehicle emission control, traffic restrictions and purchase restrictions.

To alleviate the haze pollution pressure in Beijing-Tianjin-Hebei region and surrounding cities, and promote the coordinated development of industrial energy structure, economic development, transportation and air quality, our suggestions and prospects are proposed as follows:

First, restrict the non-essential industrial use of inferior coal and inferior oil. Power industry enterprises that contribute a lot to PM_2.5_ rush to introduce low-calorie coal with low price and mix it with high-quality coal for power generation, which greatly reduces the utilization rate of coal and increases the industrial energy structure, resulting in the rise of PM_2.5_. In 2015, Chinese government issued the Interim Measures for The Management of Commodity Coal Quality. Accordingly, Beijing, Tianjin, Hebei, Shandong and Shanxi issued relevant measures to control the use of inferior coal. However, China’s relevant laws and regulations have not yet been improved, resulting in difficulties in implementation. Switzerland, Belgium and Norway have already been running the coal-free system for many years. In 2017, the UK achieved a full 24 h without burning coal for electricity, with half of its electricity coming from natural gas, a quarter from nuclear plants and the rest from wind, biomass and foreign imports. In the long run, developing clean energy technologies and coal efficiency technologies and increasing the consumption of renewable energy are far-reaching measures.

Second, abide by the standards for industrial transfer and formulate long-term development plans. Considering the spatial spillover effect of PM_2.5_, the short-term industrial transfer has only a short term effect on haze pollution control. Under the background that industrial transfer has become the inevitable requirement of industrial structure adjustment in Beijing-Tianjin-Hebei region, rational industrial planning is more necessary. As shown in Figure 3, from 2000 to 2017, Taiyuan and Yangquan in Shanxi Province always maintained a low-low spatial clusters of PM_2.5_, which was inseparable from the rational planning of industrial transfer and undertaking in Shanxi Province. As a major source of coal resources in China, Shanxi Province is also the main force to undertake industrial transfer in the eastern region. Taiyuan and Yangquan are major destinations for steel and non-ferrous metal industries. However, they can maintain low PM_2.5_ levels under the pressure of developing their own coal industry and undertaking the transfer of heavy industry from outside, which is closely related to the strict time-limited transformation or withdrawal mechanism of high-emission projects in Shanxi Province. Shanxi’s experience in dealing with emissions from the coal industry is worth learning from elsewhere, including Hebei.

Third, encourage local governments to support carless families in purchasing their first new energy vehicles. Traffic restrictions and purchase restrictions of motor vehicle can improve urban air quality in the short term. However, it reduces consumption power and has a negative impact on the development of relevant industries in the region, such as the automobile industry. It cannot be regarded as a long-term and fundamental measure of urban traffic management. Efforts should be made on the supply side to deal with traffic congestion. More sustainable governance policies include encouraging regions where conditions permit to give preferential treatment to new-energy vehicles in terms of parking fees, exploring the establishment of zero-emission zones and alleviating traffic congestion through appropriate administrative measures on traffic restrictions and purchase restrictions.

## Figures and Tables

**Figure 1 ijerph-17-05610-f001:**
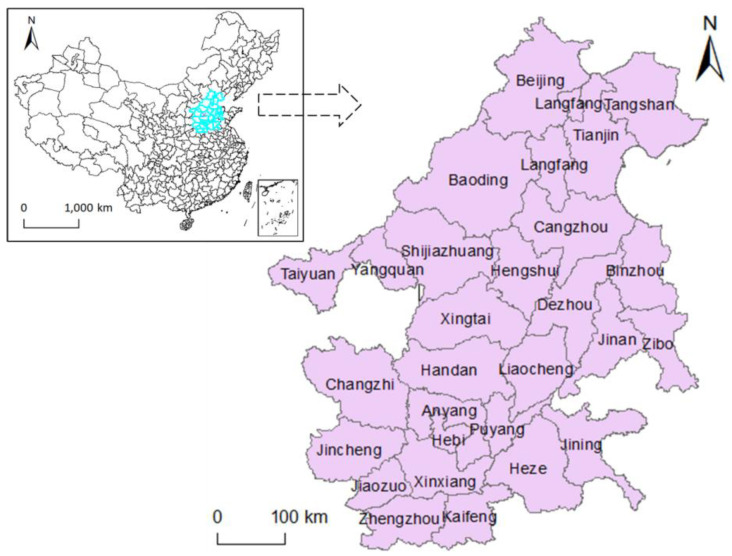
The location of Beijing-Tianjin-Hebei region and surrounding areas.

**Figure 2 ijerph-17-05610-f002:**
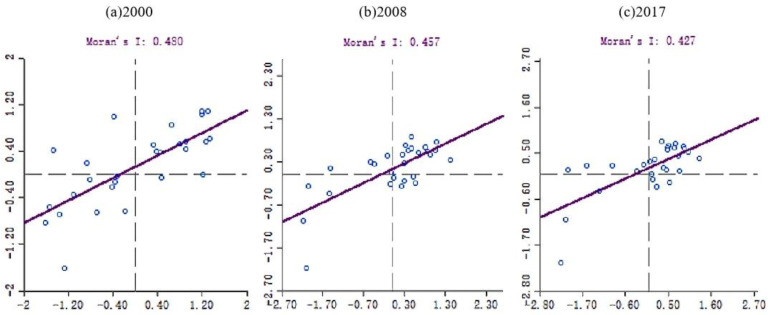
Moran scatter plot of PM_2.5_ in Beijing-Tianjin-Hebei region and surrounding cities in 2000, 2008 and 2017.

**Figure 3 ijerph-17-05610-f003:**
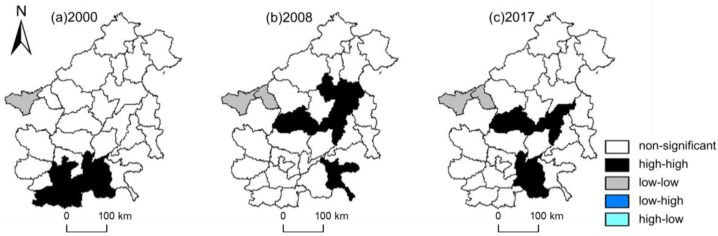
Cluster map of PM_2.5_ in Beijing-Tianjin-Hebei region and surrounding cities in 2000, 2008 and 2017.

**Figure 4 ijerph-17-05610-f004:**
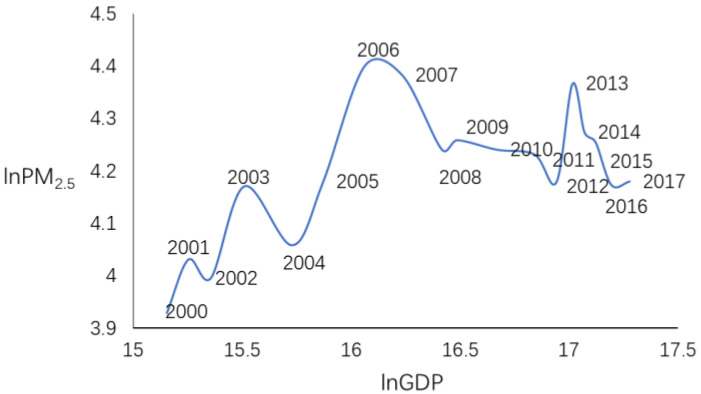
The relationship between lnGDP and lnPM_2.5_ in Beijing-Tianjin-Hebei region and surrounding cities.

**Table 1 ijerph-17-05610-t001:** Global Moran’s I for PM_2.5_ in Beijing-Tianjin-Hebei region and surrounding cities.

Year	Morans’ I	E(I)	sd(I)	Z	*p*-Value
2000	0.480	−0.037	0.143	3.613	0.001
2001	0.342	−0.037	0.135	2.802	0.002
2002	0.402	−0.037	0.142	3.074	0.001
2003	0.389	−0.037	0.132	3.240	0.001
2004	0.411	−0.037	0.134	3.316	0.001
2005	0.488	−0.037	0.141	3.724	0.001
2006	0.367	−0.037	0.135	3.033	0.001
2007	0.514	−0.037	0.140	3.932	0.001
2008	0.457	−0.037	0.136	3.612	0.001
2009	0.392	−0.037	0.135	3.163	0.001
2010	0.514	−0.037	0.137	3.953	0.001
2011	0.415	−0.037	0.138	3.281	0.001
2012	0.492	−0.037	0.139	3.817	0.001
2013	0.420	−0.037	0.138	3.266	0.001
2014	0.449	−0.037	0.137	3.544	0.001
2015	0.473	−0.037	0.137	3.705	0.001
2016	0.4355	−0.037	0.1353	3.4806	0.001
2017	0.4267	−0.037	0.137	3.4305	0.001

Note: E(I) is −1/(n-1), which is the expected value of I; Sd (I) represents the variance of the I; Z is the z test value of I, and *p*-value is its adjoint probability, which was obtained by Monte Carlo simulation for 999 permutations.

**Table 2 ijerph-17-05610-t002:** Results of spatial autoregressive model and spatial error model.

Variable	SAR			SEM		
	Model(1)	Model(2)	Model(3)	Model(4)	Model(5)	Model(6)
	Spatial Fixed Effects	Time Period Fixed Effects	Spatial and Time Period Fixed Effects	Spatial Fixed Effects	Time Period Fixed Effects	Spatial and Time Period Fixed Effects
ES	0.039 * (2.371)	0.093 *** (4.338)	0.091 *** (4.269)	0.130 *** (4.870)	0.131 *** (5.045)	0.134 *** (5.261)
lnGDP	0.026 *** (4.294)	0.046 *** (5.995)	0.043 *** (5.611)	0.054 *** (6.043)	0.057 *** (6.439)	0.055 *** (6.263)
TJ	1.083 * (2.480)	1.992 *** (3.658)	1.951 *** (3.619)	2.717 *** (4.036)	2.637 *** (4.041)	2.700 *** (4.196)
ρ	0.806 *** (37.621)	0.548 *** (13.226)	0.594 *** (15.516)			
λ				0.806 *** (36.299)	0.573 *** (13.900)	0.651 *** (18.460)
σ^2^	0.019	0.019	0.018	0.018	0.018	0.017
R^2^	0.709	0.711	0.731	0.013	0.608	0.627
LM(lag)	302.732 ***	44.031 ***	37.833 ***	475.031 ***	93.809 ***	92.769 ***
R-LM(lag)	3493.336 ***	34.415 ***	36.036 **	76.712 ***	0.545	0.920
LM(error)	23.404 ***	22.422 ***	17.515 ***	628.315 ***	121.273 ***	122.694 ***
R-LM(error)	3214.008 ***	12.807 ***	15.718 ***	229.995 ***	28.010 ***	30.845 ***

Note: *, **, and *** respectively indicate that the estimated coefficient is significant at *p* < 0.05, *p* < 0.01 and *p* < 0.001. The value in parentheses is the t value of the coefficient.

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
