# Peer review of "Spatial Effect of Industrial Energy Consumption Structure and Transportation on Haze Pollution in Beijing-Tianjin-Hebei Region"

_ijerph, 2020, doi:10.3390/ijerph17155610_

Round 1
Reviewer 1 Report
This is a well-written paper. The methodology is clearly described.
Minor comments and suggestions for the authors to consider:
- Line 22: define EKC (environmental Kuznets curve) in the abstract
- Line 64: “of which the contribution of automobile exhaust is about 10% to 30%”. I would expect that the contributions from coal burning sources, industrial sources and automobile exhaust could vary with the locations and the season. So the citation of 10-30 % should be clarified.
- Line 73: “transport”: should be “transportation”
- Line 77: “so as to promote the further development of regional comprehensive pollution control.”. I suggest that the authors strengthen the justification for this study. The study area is briefly described in section 2. However, the introduction provides minimal background information on this study area. The Beijing-Tianjin-Hebei region is among the most polluted areas for China. There are extensive studies for this area using various approaches. Therefore, what unique and/or new perspectives that this study is aimed at should be addressed.
- Line 187: Figure 1 should be Figure 2.
- Lines 174-182 (Moran scatter plot): The figure panels are for 2000, 2008, 2017. (i) It is not clear to me why these 3 years are chosen here. (ii) This paper is focused on spatial effect. However, the key factors considered here do have temporal variations. Meteorological conditions that could affect the formation/distribution/removal of PM2.5 are also exhibit year-to-year variations. These factors should be addressed.
- Line 202: Figure 2 should be Figure 3
- Lines 310: “two aspects of energy conservation and emission reduction, industrial transfer”: from the aspects of energy conservation and emission reduction, industrial transfer?
- Lines 313: “ two aspects of vehicle emission control, traffic restrictions and purchase restrictions. “: See previous comment.
- Line 334: “Figure 2”. It is Figure 3, I think.
Author Response
We acknowledge the reviewer’s comments and suggestions very much, which are valuable in improving the quality of our manuscript.
Dear reviewer,
I am very grateful to your comments for the manuscript. According with your advice, we amended the relevant part in manuscript. Some of your questions were answered below.
Point 1: Line 22: define EKC (environmental Kuznets curve) in the abstract
Response 1: Please refer to lines 288-289 for the definition of EKC.
Point 2: Line 64: “of which the contribution of automobile exhaust is about 10% to 30%”. I would expect that the contributions from coal burning sources, industrial sources and automobile exhaust could vary with the locations and the season. So the citation of 10-30 % should be clarified.
Response 2: Thanks for the reviewer's recommendation. The citation of “10-30%” has been added.(line 83)
Point 3: Line 73: “transport”: should be “transportation”.
Response 3: Thanks for the reviewer's recommendation. Please refer to line 92 for the modification.
Point 4: Line 77: “so as to promote the further development of regional comprehensive pollution control.”. I suggest that the authors strengthen the justification for this study. The study area is briefly described in section 2. However, the introduction provides minimal background information on this study area. The Beijing-Tianjin-Hebei region is among the most polluted areas for China. There are extensive studies for this area using various approaches. Therefore, what unique and/or new perspectives that this study is aimed at should be addressed.
Response 4: There are two new perspectives in this study. First, “traffic jam” was regarded as a factor to analyze the spatial influence. Second, most of the existing studies chose the "Beijing-Tianjin-Hebei City Cluster", namely the original 14 cities, as the study area. The 28 cities chosen in this study are the heavily polluted area redefined after 2016, with a wider and more accurate coverage. This explanation has been added into the revised manuscript.(lines 95-99)
Point 5: Line 187: Figure 1 should be Figure 2.
Response 5: Thanks for the reviewer's correction. Please refer to line 216 for the modification.
Point 6: Lines 174-182 (Moran scatter plot): The figure panels are for 2000, 2008, 2017. (i) It is not clear to me why these 3 years are chosen here. (ii) This paper is focused on spatial effect. However, the key factors considered here do have temporal variations. Meteorological conditions that could affect the formation/distribution/removal of PM2.5 are also exhibit year-to-year variations. These factors should be addressed.
Response 6: (i) In fact, it can be seen from Table 1 that each year from 2000 to 2017 has been analyzed in this part. However, due to limited space, all the 18 corresponding Moran scatter plots cannot be shown. Therefore, the representative results of 2000, 2008 and 2017 were selected to illustrate the changing trend. (ii) It is true that meteorological conditions have an impact on atmospheric conditions, but it is not consistent with the perspective of this study. The perspective of this study is the combination of environmental science and economics. The selected influencing factors are all related to social and economic development, but meteorological factors are pure natural science factors, so they are not suitable to be included in this study.
Point 7: Line 202: Figure 2 should be Figure 3
Response 7: Thanks for the reviewer's correction. Please refer to line 231 for the modification.
Point 8: Lines 310: “two aspects of energy conservation and emission reduction, industrial transfer”: from the aspects of energy conservation and emission reduction, industrial transfer?
Response 8: These two aspects are: (i) Energy conservation and emission reduction; (ii) Industrial transfer. "Energy conservation and emission reduction" is a professional term in the fields of economics and environmental science. It is a whole and cannot represent two parts.
Point 9: Lines 313: “two aspects of vehicle emission control, traffic restrictions and purchase restrictions.”: See previous comment.
Response 9: Thanks for the reviewer's correction. The description here is indeed misleading. Please refer to line 345 for the modification.
Point 10: Line 334: “Figure 2”. It is Figure 3, I think.
Response 10: Thanks for the reviewer's correction. Please refer to line 366 for the modification.
We acknowledge the reviewer’s comments and suggestions very much, which are valuable in improving the quality of our manuscript.
Kindest regards,
Meicun Li
Reviewer 2 Report
This paper analyzes how industrial energy consumption structure and transportation affect haze pollution in Beijing-Tianjin-Hebei region. This paper systematically studies the issues to be explored. However, the paper has some deficiencies which I list below in order of significance. Let me be more specific with my decision.
1. Authors ought to spend more time to check full manuscript to avoid all kinds of errors. For example, there are two ‘Fig1’ in this paper. In section 3.1, which picture is Figure 3 in ‘As shown in Figure 3’?
2. It is suggested that the authors improve the clarity of the pictures in this paper.
3. Section 1 includes Introduction and Literature review, but the contents are limited. In my opinion, there is a need to enrich this section. For example, the authors can present air pollution conditions (10.1016/j.jclepro.2019.03.152) and environment pollution status (10.1016/j.jclepro.2019.118960) in China. In addition, it is important to discuss the determinants (10.1016/j.jclepro.2019.03.105) and reduction mechanism of haze pollution (10.1016/j.jclepro.2019.118889) in Literature review. In this way, Section 1 could be enriched and improved. The referred papers are as follows.
Examining the synergistic effect of CO2 emissions on PM2.5 emissions reduction: Evidence from China. J. Clean. Prod. 223: 759-771. (10.1016/j.jclepro.2019.03.152)
Can industrial agglomeration promote pollution agglomeration? Evidence from China. J. Clean. Prod. 246: 118960. (10.1016/j.jclepro.2019.118960)
Determinants of haze pollution: An analysis from the perspective of spatiotemporal heterogeneity. J. Clean. Prod. 222: 768-783. (10.1016/j.jclepro.2019.03.105)
Causal chain of haze decoupling efforts and its action mechanism: Evidence from 30 provinces in China. J. Clean. Prod. 245: 118889. (10.1016/j.jclepro.2019.118889)
4. The literature review section of this paper is too general and does not systematically review the theoretical basis. Moreover, the innovation of this paper is not clear.
5. The chapter title of this paper does not match the content. For example, section 4 is discussion and conclusion, but this part does not reflect the discussion.
6. There are some English writing problems in this manuscript from my reading. It is suggested that the authors find a native English speaker to improve the English writing of this manuscript.
Author Response
Please see the attachment.
We acknowledge the reviewer’s comments and suggestions very much, which are valuable in improving the quality of our manuscript.

Reviewer 3 Report
This study exploring the spatial effect of industrial energy consumption structure and traffic factors on haze pollution by using various source dataset, spatial autoregressive model (SAR) and spatial error model (SEM) model. This study has well structure and quantitative evaluation methods. However, the research context, theoretical basis and significance of this study are still insufficient. The necessity of analytical methods’ use is not well explained, and the value of the research results needs to be further summarized. Therefore, this paper is not suitable for publication in its current form. Here are some comments or suggestions for improving the current version.
- The author does not provide the detailed explanation of methods of use. For example, the spatial autoregressive model (SAR) and spatial error model (SEM) model in the paper, why use these methods rather than other regression models (e.g., Ordinary least squares and geographically weighted regression)? Which studies have used these methods in this field before? What are the advantages and disadvantages of these methods for the research questions?
- The author needs to give reasons for using the spatial autoregressive model (SAR) and spatial error model (SEM) to analyze spatial effect of industrial energy consumption structure and traffic factors on haze pollution. At the same time, a corresponding literature review is also required.
- This paper lacks a more detailed analysis of the validity and credibility of the traffic jam. The transportation variable (i.e., traffic jam) is defined as the ratio of the consumption of gasoline and diesel oil of urban residents to the urban road area(t/m2). The data for estimating this index is from Statistical Yearbook(2001-2018) and Statistical Communique on National Economic and Social Development(2000-2017), which has aggregated into each city. Does the index enough to capture a degree of the traffic jam for a city without considering any spatial variations effect? The authors should present the detailed description of variables before result section.
- In the research results, there are many inferences about the reasons, which are more suitable in the discussion section of the article.
- Some sentences are incomplete, and some have grammatical problems. For example, in line 59 – 60, ‘Chen et al. [1],Adgate et al. [21], Liu et al. [22] 59 and Bell et al. [23]pointed out the spatial and temporal characteristics of the distribution of PM2.5’. What kind of spatial and temporal characteristics of the distribution of PM2.5 these studies have pointed out? The language in this paper needs revision to ensure that the readers are able to fully understand the contributions of this paper.
Author Response
We acknowledge the reviewer’s comments and suggestions very much, which are valuable in improving the quality of our manuscript.
Dear reviewer,
I am very grateful to your comments for the manuscript. According with your advice, we amended the relevant part in manuscript. Some of your questions were answered below.
Point 1: The author does not provide the detailed explanation of methods of use. For example, the spatial autoregressive model (SAR) and spatial error model (SEM) model in the paper, why use these methods rather than other regression models (e.g., Ordinary least squares and geographically weighted regression)? Which studies have used these methods in this field before? What are the advantages and disadvantages of these methods for the research questions?
Response 1: Thanks for the reviewer's recommendation. Spatial econometric panel data model was developed by the cross-section data of multiple time nodes. Anselin (1988) introduced spatial effect into the cross-section data model, and proposed Spatial Autoregression Model (SAR) and Spatial Error Model (SEM) .Anselin et al. (1996) proposed the Ordinary Least Squares (OLS) for the benchmark models of SAR and SEM, and developed the Lagrangian Multiplier (LM) statistic to test the autocorrelation of the spatial lag term and the spatial error term. Using spatial statistical models with fixed effects items and random effects items, spatial econometric panel data model controls the individual heterogeneity. This explanation has been added into the revised manuscript.(lines 43-49)
Point 2: The author needs to give reasons for using the spatial autoregressive model (SAR) and spatial error model (SEM) to analyze spatial effect of industrial energy consumption structure and traffic factors on haze pollution. At the same time, a corresponding literature review is also required.
Response 2: Thanks for the reviewer's recommendation. Hosseini and Kaneko(2013) used SDM to study the atmospheric conditions of 129 countries, and proved that pollutants could spread to neighboring countries, and a significant inverted U-shaped EKC curve could be seen after controlling the spatial relationship. Ding et al. (2019) adopted SDM to test whether there was EKC trend between PM2.5 and economic growth in Beijing-Tianjin-Hebei region, and the results showed that economic growth and PM2.5 presented an obvious inverted U-shaped EKC curve. Compared with the non-spatial model, the turning point of EKC is more likely to occur when considering the spatial effect. Ma et al. (2014) used SEM and SAR to analyze the interaction effect of inter-provincial PM10 and the influence of industrial structure, and found that industrial transfer lacked long-term impact on reducing pollution. Taking Beijing as an example, they found that traffic jam and spatial factors were important reasons for severe pollution. This explanation has been added into the revised manuscript.(lines 65-75)
Point 3: This paper lacks a more detailed analysis of the validity and credibility of the traffic jam. The transportation variable (i.e., traffic jam) is defined as the ratio of the consumption of gasoline and diesel oil of urban residents to the urban road area(t/m2). The data for estimating this index is from Statistical Yearbook(2001-2018) and Statistical Communique on National Economic and Social Development(2000-2017), which has aggregated into each city. Does the index enough to capture a degree of the traffic jam for a city without considering any spatial variations effect? The authors should present the detailed description of variables before result section.
Response 3: In 2006, the Intergovernmental Panel on Climate Change (IPCC) published the relationship between the speed of private cars and the amount of petrol consumed per mile in the Guidelines for National Greenhouse Gas Inventory. It pointed out that the slower a car goes, the more petrol it consumes. In traffic jams, the petrol consumption is almost twice that of normal driving. Ma et al.(2014) took traffic congestion as an independent variable and carried out a spatial analysis of the influencing factors of haze pollution, and found that the indicator of traffic jam was not significant in whole China. This study made a spatial analysis of Beijing-Tianjin-Hebei region and surrounding cities to further explore the impact of traffic jam on haze pollution. This explanation has been added into the revised manuscript.(lines 184-188)
Point 4: In the research results, there are many inferences about the reasons, which are more suitable in the discussion section of the article.
Response 4: Thanks for the reviewer's recommendation. In the results section, in addition to the description of the results, the causes of the results were briefly analyzed, while the discussion section focused more on the analysis of ways to mitigate pollution.
Point 5: Some sentences are incomplete, and some have grammatical problems. For example, in line 59-60, ‘Chen et al. [1],Adgate et al. [21], Liu et al. [22] 59 and Bell et al. [23]pointed out the spatial and temporal characteristics of the distribution of PM2.5’. What kind of spatial and temporal characteristics of the distribution of PM2.5 these studies have pointed out? The language in this paper needs revision to ensure that the readers are able to fully understand the contributions of this paper.
Response 5: Thanks for the reviewer's correction. Please refer to line 78 for the modification.
We acknowledge the reviewer’s comments and suggestions very much, which are valuable in improving the quality of our manuscript.
Kindest regards,
Meicun Li
Reviewer 4 Report
The manuscript entitled “Spatial Effect of Industrial Energy Consumption Structure and Transportation on Haze Pollution in Beijing-Tianjin-Hebei Region” presents an interesting assessment of several determinants on particulate matter in a severe polluted area of China. The manuscript is in general clear and well-written. Some revisions are suggested in order to improve the general good quality of the research.
Please carefully revised the Figure numbering since Figure 1 is reported twice and numbering is not consistent between text and figures. Also in Figure 2 at page 202, since some spatial cluster are not present, figure legend could be revised.
A brief description of stages meaning of national emission standards could be provided in order to understand differences and improvement from lower to higher stages also outside Chinese readers probably already used to their meaning.
Please provide a reference for statement at page 11, lines 317-320. Also in general in the Discussion, some statements lack of a proper citation. Please check and revise.
At page 1, lines: please check page numbers reported
Please check formatting of Table 2, it should be improved. Also, please add a list of all abbreviations in the footnote.
Author Response
Please see the attachment.
We acknowledge the reviewer’s comments and suggestions very much, which are valuable in improving the quality of our manuscript.
Dear reviewer,
I am very grateful to your comments for the manuscript. According with your advice, we amended the relevant part in manuscript. Some of your questions were answered below.
Point 1: Please carefully revised the Figure numbering since Figure 1 is reported twice and numbering is not consistent between text and figures. Also in Figure 2 at page 202, since some spatial cluster are not present, figure legend could be revised.
Response 1: Thanks for the reviewer's correction. The numbers of figures have been modified.
Point 2: A brief description of stages meaning of national emission standards could be provided in order to understand differences and improvement from lower to higher stages also outside Chinese readers probably already used to their meaning.
Response 2: Thanks for the reviewer's recommendation. China's emission standard is the same as that in Europe. Please refer to line 316 for the modification.
Point 3: Please provide a reference for statement at page 11, lines 317-320. Also in general in the Discussion, some statements lack of a proper citation. Please check and revise.
Response 3: Thanks for the reviewer's recommendation. Since the statements in the discussion section were based on the results section, it is not appropriate to add some other citations here.
Point 4: At page 1, lines: please check page numbers reported
Response 4: Thanks for the reviewer's correction.
Point 5: Please check formatting of Table 2, it should be improved. Also, please add a list of all abbreviations in the footnote.
Response 5: Thanks for the reviewer's correction. Table 2 has been improved. The abbreviations were explained in lines 162-163.
We acknowledge the reviewer’s comments and suggestions very much, which are valuable in improving the quality of our manuscript.
Kindest regards,
Meicun Li
Round 2
Reviewer 2 Report
All my concerns have been addressed.
Reviewer 3 Report
The authors have addressed my concerns, and I don't have any further comments for the time being. Extensive English editing is required before publication.
Reviewer 4 Report
The manuscript has been improved from the first draft.